# National analysis of cancer mortality and proximity to nuclear power plants in the United States

Yazan Alwadi [1] ✉, Barrak Alahmad [1], Carolina L. Zilli Vieira [1], Philip J. Landrigan[2,3], David C. Christiani [1,4,5], Eric Garshick[5,6], Marco Kaltofen [7], Brent Coull[1,8], Joel Schwartz [1], John S. Evans [1] & Petros Koutrakis[1]

Understanding the potential health implications of living near nuclear power plants is important given the renewed interest in nuclear energy as a low-carbon power source. Here we show that U.S. counties located closer to operational nuclear power plants have higher cancer mortality rates than those farther away. Using nationwide mortality data from 2000-2018, we assess long-term spatial patterns of cancer mortality in relation to proximity to nuclear facilities while accounting for socioeconomic, demographic, behavioral, environmental, and healthcare factors. Cancer mortality is higher across multiple age groups in both males and females, with the strongest associations among older adults, males aged 65–74 and females aged 55–64. While our findings cannot establish causality, they highlight the need for further research into potential exposure pathways, latency effects, and cancer-specific risks, emphasizing the importance of addressing these potentially substantial but overlooked risks to public health.

Nuclear power plants have long been a major source of energy production worldwide, playing a critical role in electricity generation. As of 2023, approximately 440 nuclear reactors were operational globally, with a combined capacity of about 390 gigawatts electrical (GWe), generating 2602 terawatt-hours (TWh) of electricity and accounting for approximately 9% of global electricity production. With about 60 additional reactors under construction and over 110 planned, nuclear energy remains an important source. In addition, approximately 30 countries are considering or initiating nuclear power programs (World Nuclear Association (WNA)).

The United States began generating electricity from commercial nuclear power plants in 1958 and is now the world's largest producer of nuclear energy, contributing about 30% of global nuclear electricity. As

of August 1, 2023, the U.S. operates 93 commercial nuclear reactors across 54 plants in 28 states, providing a significant portion of the nation's electricity. The average reactor age is approximately 42 years, reflecting the long-term reliance on nuclear energy for power generation (WNA, U.S. Energy Information Administration (EIA)).

Nuclear power plants emit radioactive pollutants that can disperse into the surrounding environment, leading to potential human exposure through inhalation, ingestion, and direct contact. These pollutants can be transported through air, water, and soil, contributing to long-term environmental contamination[1]. Populations residing near nuclear power plants may experience low-level chronic exposure to ionizing radiation via environmental release pathways. While our study does not include dosimetry, ionizing radiation is a well-established

[1]Environmental Health Department, Harvard T.H. Chan School of Public Health, Boston, MA, USA. [2]Boston College, Chestnut Hill, MA, USA. [3]Centre Scientifique de Monaco, Monaco, Monaco. [4]Division of Pulmonary and Critical Care Medicine, Department of Medicine, Massachusetts General Hospital, Boston, MA, USA. [5]Harvard Medical School, Boston, MA, USA. [6]Pulmonary, Allergy, Sleep, and Critical Care Medicine Section, Medical Service, VA Boston Healthcare System, Boston, MA, USA. [7]Boston Chemical Data Corp, Natick, MA, USA. [8]Department of Biostatistics, Harvard T.H. Chan School of Public Health, Boston, MA, USA. ✉e-mail: yazan_alwadi@fas.harvard.edu; Yazan.alwadi.epi@gmail.com

carcinogen[2–7] and thus motivates investigation into proximity-based exposure patterns. Given these concerns, numerous studies have examined whether living near nuclear power plants is associated with an increased risk of cancer, but their findings have been inconsistent.

Globally, some studies have reported no association between proximity to nuclear power plants and increased cancer risk[8–15]. While other studies have found significant associations between residential proximity to nuclear power plants and increased cancer incidence[16–20]. The conflicting nature of these findings underscores the need for further investigation into the potential health effects of nuclear power plant emissions.

Despite the importance and prevalence of nuclear power plants in the U.S., epidemiologic research regarding their health impacts remains rare. Most U.S. studies have focused on individual plants or limited regions, with only a few national assessments to date - many of which relied on fixed distance cutoffs to classify exposed populations[8,9,11,12,19,21–25]. These studies often focus on a single facility and its surrounding communities, which restricts their statistical power to detect effects and ability to capture broader exposure patterns. Furthermore, differences in study design, exposure assessment methods, and geographic scope make it difficult to draw generalizable conclusions.

In this work, we assess the association between county-level proximity to nuclear power plants and cancer mortality across the United States from 2000 to 2018. We find that counties located closer to operational nuclear power plants have higher cancer mortality rates, with stronger associations observed among older adults. These associations remain consistent across multiple sensitivity analyses and proximity definitions. The results highlight spatial patterns of cancer risk in relation to nuclear power generation and emphasize the importance of evaluating potential long-term health implications of nuclear energy infrastructure in population-scale studies.

## Results
### Nuclear power plants proximities
Figure 1 presents the county-level nuclear power plants proximity estimates for the year 2000, illustrating the 10-year average proximity of the sum of inverse-distance (in kilometers) nuclear power plants proximity to operational nuclear power plants. Proximity was calculated by summing the inverse-distance weights from all nuclear plants within 200 km of each county center.

Counties closer to multiple nuclear power plants had higher estimated proximity levels, as indicated by the darker shading on the map. While distance from the nearest plant played a significant role, cumulative proximities from multiple plants also contributed to variation across counties. Counties in the Midwest, Northeast, and parts of the Southeast exhibited the highest nuclear power plants proximity levels, while regions in the West and Great Plains had lower proximities due to the sparser distribution of nuclear power plants.

Figure 2 presents the cumulative population distribution by nuclear power plants proximity level and its distance-equivalent representation. Proximity is calculated as the sum of inverse distances ($1/d$) from all operational nuclear power plants within 200 km, while the equivalent distance represents the inverse of this sum, effectively translating cumulative nuclear power plant proximity into the distance from a single nuclear power plant that would result in the same proximity level.

This figure highlights the number of individuals residing at or below each level of nuclear power plants proximity, or equivalently, within a comparable distance of a single operational nuclear power plant.

### Cancer mortality
Figure 3 presents the estimated associations between nuclear power plants proximity and cancer mortality, showing the coefficients and 95% confidence intervals (CIs) for each age group and sex. The results indicate a positive association between nuclear power plants proximity and cancer mortality, with the highest relative risk observed in the 55–64 age group for females and 10 years later (65–74) in males.

Table 1 summarizes the total number of cancer deaths attributable to nuclear power plants proximity, stratified by age group and sex. The estimated number of attributable cancer deaths was lowest in the 35–44 age group, with 591 (95% CI: −538, 1696) for females and 260 (95% CI: −753, 1248) for males. The burden increased progressively with age, peaking in the 65–74 age group for females (13,976; 95% CI: 6885, 20,959) and the 65–74 age group for males (20,912; 95% CI: 12,591, 29,109).

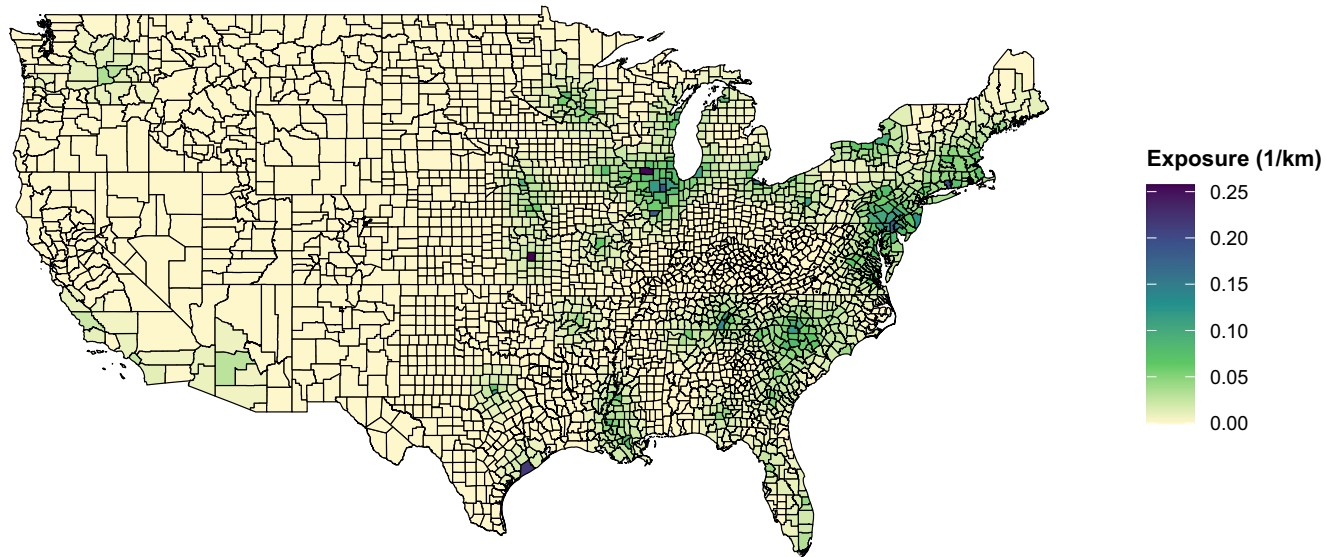

**Fig. 1 | County-level 10-year average proximity to nuclear power plants (1/km) in the United States, year 2000.** Map showing the spatial distribution of aggregated inverse-distance proximity (1/km) to operational nuclear power plants for each U.S. county in the year 2000. Darker shades (purple) indicate higher cumulative proximity from multiple plants within 200 km, and lighter shades (yellow) indicate lower proximity. County boundaries are outlined in black. Source data are provided as a Source Data file.

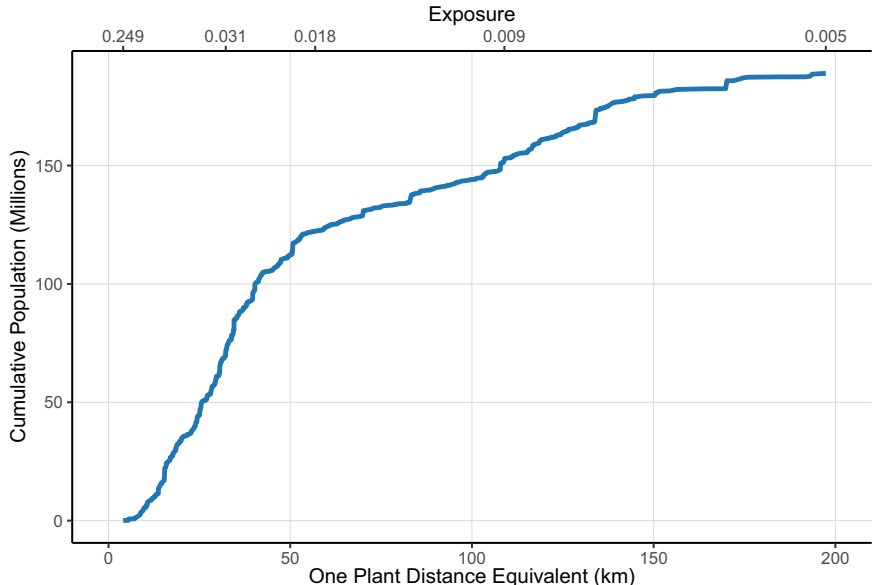

**Fig. 2 | Cumulative population exposed to each level of nuclear power plant proximity or greater, or residing at an equivalent distance or closer to an operational nuclear power plant, 2018.** Plot showing the cumulative U.S. population (in millions) living within increasing proximity levels (sum of inverse distances) or equivalent single-plant distances from operational nuclear power plants in 2018. The solid blue line represents the cumulative population curve, with axes labeled for both proximity (top axis) and equivalent distance (bottom axis). Source data are provided as a Source Data file.

| Female | Age | Estimate (CI) | |
|---|---|---|---|
| | 35_44 | 0.22 (−0.19, 0.63) | |
| | 45_54 | 0.55 (0.22, 0.88) | |
| | 55_64 | 0.74 (0.45, 1.03) | |
| | 65_74 | 0.56 (0.27, 0.85) | |
| | 75_84 | 0.42 (0.18, 0.66) | |
| | 85+ | 0.48 (0.21, 0.75) | |
| **Male** | | | |
| | 35_44 | 0.13 (−0.36, 0.61) | |
| | 45_54 | 0.36 (−0.00, 0.73) | |
| | 55_64 | 0.61 (0.30, 0.91) | |
| | 65_74 | 0.70 (0.42, 0.99) | |
| | 75_84 | 0.57 (0.34, 0.81) | |
| | 85+ | 0.40 (0.14, 0.66) | |

**Fig. 3 | Estimated associations between nuclear power plant proximity and cancer mortality by age group and sex, 2000–2018.** Forest plot presenting regression coefficients and 95% confidence intervals (CIs) for associations between county-level proximity to nuclear power plants and cancer mortality, stratified by age group and sex. Estimates were derived from generalized estimating equation (GEE) Poisson regression models with a log link, incorporating a population offset and robust (sandwich) variance estimators. Confidence intervals are based on two-sided statistical tests. Each observation represents an independent county–year–age–sex unit ($n \approx 290{,}000$). Data represent independent population units, not biological or technical replicates. Horizontal bars indicate 95% CIs around model-estimated effects. Source data are provided as a Source Data file.

Among females, the second highest number of attributable deaths occurred in the 55–64 age group (13,070; 95% CI: 8057, 18,008), followed by the 75–84 age group (12,397; 95% CI: 5383, 19,321). The 85+ age group had the lowest burden among older females, with 9451 (95% CI: 4146, 14,678).

Among males, the second highest number of attributable deaths was in the 75–84 age group (17,968; 95% CI: 10,599, 25,243), followed by the 55–64 age group (12,611; 95% CI: 6380, 18,745). The 85+ age group had the lowest burden among older males, with 6341 (95% CI: 2274, 10,352).

Figure 4 illustrates the relative risk of cancer mortality by equivalent plant distance, stratified by sex and age group.

These plots demonstrate a consistent inverse association between distance from a nuclear plant and cancer mortality risk, with the highest relative risks observed at shorter distances and a gradual decline as distance increases. The highest relative risk among females was 1.19 in the 65–74 age group, while the highest relative risk among males was 1.20 in the 65–74 age group. The cumulative population (millions) displayed on the secondary x-axis contextualizes the

**Table 1 | Estimated attributable cancer mortality due to proximity to nuclear power plants, stratified by age group and sex, United States, 2000–2018**

| Age Group | Sex | Attributable Cancer Mortality (95% CI) | Total Cancer Mortality | Attributable Fraction (CI) |
|---|---|---|---|---|
| 35–44 | Female | 591 (−538, 1696) | 97,536 | 0.6% (−0.6%, 1.7%) |
| 45–54 | Female | 4864 (1945, 7732) | 311,442 | 1.6% (0.6%, 2.5%) |
| 55–64 | Female | 13,070 (8057, 18,008) | 624,238 | 2.1% (1.3%, 2.9%) |
| 65–74 | Female | 13,976 (6885, 20,959) | 878,866 | 1.6% (0.8%, 2.4%) |
| 75–84 | Female | 12,397 (5383, 19,321) | 1,010,316 | 1.2% (0.5%, 1.9%) |
| 85+ | Female | 9451 (4146, 14,678) | 670,248 | 1.4% (0.6%, 2.2%) |
| 35–44 | Male | 260 (−753, 1248) | 73,665 | 0.4% (−1%, 1.7%) |
| 45–54 | Male | 3145 (−5, 6235) | 307,074 | 1% (0%, 2%) |
| 55–64 | Male | 12,611 (6380, 18,745) | 746,782 | 1.7% (0.9%, 2.5%) |
| 65–74 | Male | 20,912 (12,591, 29,109) | 1,062,167 | 2% (1.2%, 2.7%) |
| 75–84 | Male | 17,968 (10,599, 25,243) | 1,105,497 | 1.6% (1%, 2.3%) |
| 85+ | Male | 6341 (2274, 10,352) | 559,185 | 1.1% (0.4%, 1.9%) |
|  | **Total** | **115,586 (56,964–173,326)** |  |  |

The table presents estimated numbers of cancer deaths attributable to residential proximity to nuclear power plants, stratified by age group and sex, based on generalized estimating equation (GEE) Poisson regression models. Values are reported as point estimates with 95% confidence intervals (CIs). CIs were calculated using two-sided tests based on robust (sandwich) variance estimators. The attributable fraction (AF) represents the percentage of total cancer mortality within the studied counties estimated to be attributable to nuclear power plant proximity within each stratum, with corresponding 95% CIs.

population distribution at different risk levels, highlighting the number of individuals affected at varying distances.

These results suggest that living near nuclear power plants is associated with increased cancer mortality risk, particularly in older populations. The estimated cancer burden attributable to nuclear power plants proximity underscores the potential public health implications of residential proximity to operational nuclear power plants.

We conducted sensitivity analyses to assess the robustness of our findings. First, we reran our models varying the distances to county centers from 200 km down to 100 km, in increments of 10 km, and the results remained consistent. Additionally, we varied the average proximity windows across intervals of 2, 5, 10, 15, and 20 years, with results remaining stable. These analyses confirm that our findings are not driven by arbitrary choices in model variables or parameters, thereby reinforcing the validity of the observed associations.

## Discussion

We assessed the relationship between long-term proximity to nuclear power plants and cancer mortality using a spatially resolved, inverse-distance weighted proximity metric that captures cumulative contributions from multiple plants within 200 km of each U.S. county center. We observed positive associations between proximity and cancer mortality, with stronger effects in older age groups (Fig. 3). The highest attributable cancer mortality burden was observed among individuals aged 65–74 and 75–84, particularly among males (Table 1), reflecting both greater vulnerability in older populations and latency patterns typical of environmentally related cancers. Overall, cancer mortality among individuals aged 65 and older, associated with proximity to nuclear power plants, averaged 4266 deaths per year (95% CI: 3000–9122) between 2000 and 2018.

To contextualize these findings, we compared our results to estimates of mortality associated with another major energy source—coal. A recent study estimated that total all-cause mortality attributable to coal-fired power plant emissions averaged 20,909 deaths per year (95% CI: 19,091–22,727) over 22 years (1999–2020)[26]. While this is not a direct comparison—since our study examines cancer-caused mortality, whereas the coal study estimates all-cause mortality—our findings represent approximately 20% of the total coal-attributable deaths in their study. This comparison underscores the magnitude of cancer burden within the broader landscape of energy-related health risks.

Ionizing radiation is a well-established carcinogen, with extensive epidemiologic evidence linking radiation exposure to increased cancer risk[6,7,27]. Some of the strongest evidence comes from studies of nuclear disasters, where high-dose radiation exposure has been consistently associated with increased cancer incidence.

The Japanese atomic bomb survivor Life Span Study first identified excess leukemia deaths, which were observed were first observed about 2 years after the bombing and continued to be seen for the next 25 years after the disaster. Over time, elevated risks of solid cancers emerged, including cancers of the esophagus, stomach, colon, liver, lung, and urinary bladder, as well as female breast, brain, thyroid, and non-melanoma skin cancers. These cancers were first observed 10 years after the bombing and persisted throughout the lifetime of the cohort[28–31].

Epidemiologic findings from Ukraine following the Chernobyl disaster in 1986 have been inconsistent. A study investigating solid tumor trends 30 years after the disaster found increased incidences of colon, rectal, kidney, thyroid, breast (in women), and prostate (in men) cancers, although decreases in lung and stomach cancers were also reported[32]. In contrast, another study suggested that post-Chernobyl cancer incidence trends largely mirrored pre-disaster trends, implying no significant excess risk due to radiation exposure[33].

Studies investigating the effects of nuclear power plants on cancer incidence in surrounding communities have reported mixed findings. A study in France examining proximity to nuclear power plants and 12 cancer types found an excess incidence of bladder cancer in both males and females[20]. Similarly, a study in Spain investigating the Trillo and Zorita power plants reported that the risk of all cancers increased linearly with proximity to the plants[17]. However, other large national studies, including those conducted in the United States[8,9,25], the United Kingdom[15], and Canada[1], have found no consistent associations between residential proximity to nuclear power plants and overall or site-specific cancer incidence or mortality.

A German study focusing on pediatric cancers found a statistically significant risk ratio (RR) of 2.2 for leukemia and an RR of 1.6 for solid tumors among children under five years old living within 5 km of nuclear power plants compared to those living further away[34].

The relationship between nuclear power plants and cancer risk has also been examined in South Korea, Canada, and China, with conflicting results. A study in South Korea found elevated cancer incidence in populations near a nuclear power plant, including thyroid cancer (HR: 3.38 in females, 1.74 in males), female breast cancer (HR:

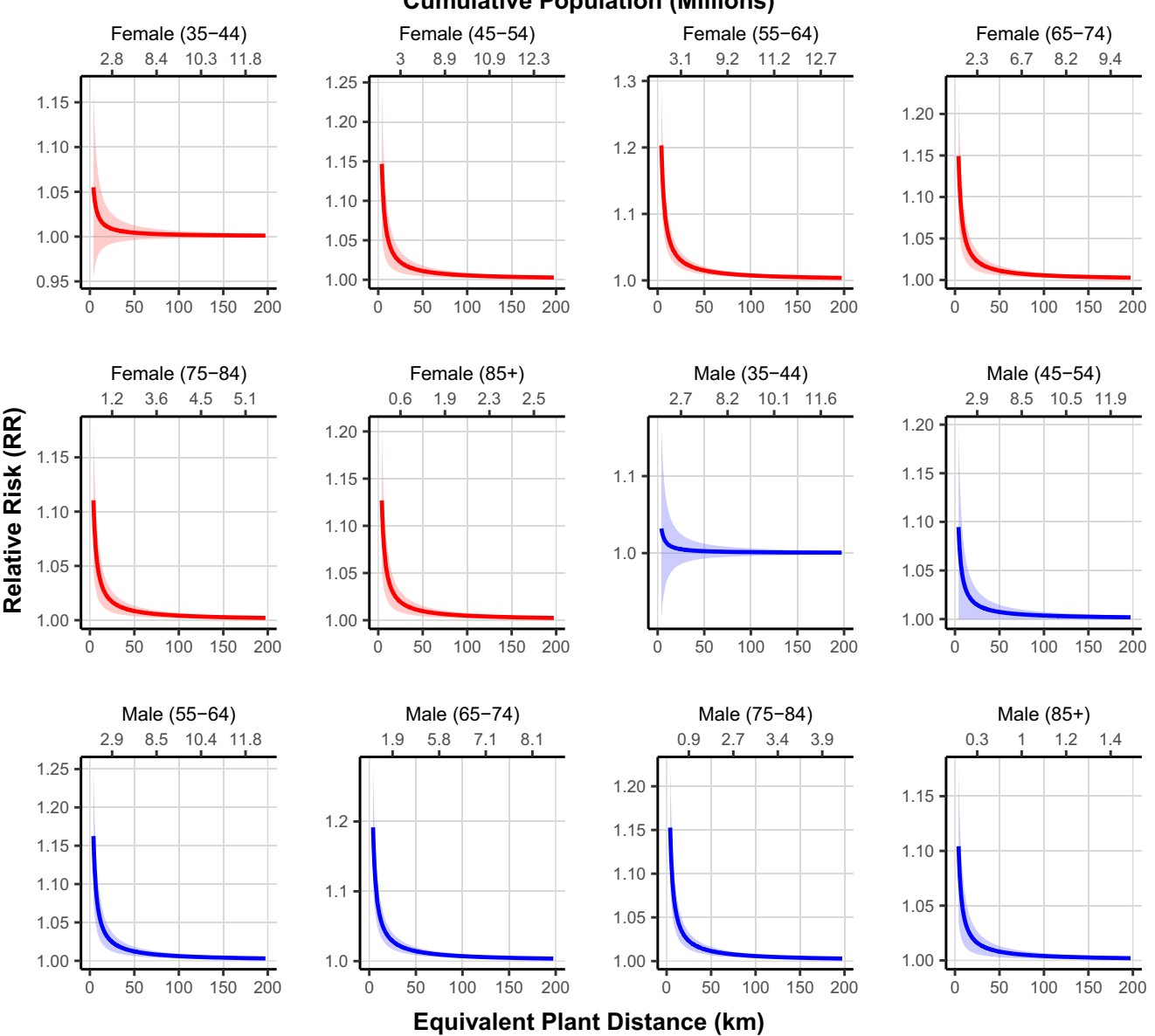

**Fig. 4 | Relative risk of cancer mortality by equivalent plant distance and cumulative population living under such risks for males and females across age groups, 2018.** Curves show model-predicted relative risks (RR) of cancer mortality as a function of equivalent plant distance, stratified by sex and age group. RRs and 95% confidence intervals (shaded bands) were estimated using GEE Poisson regression models including demographic, socioeconomic, behavioral, and environmental covariates. Each observation represents an independent county–year–age–sex unit ($n \approx 290{,}000$). Source data are provided as a Source Data file.

2.24), and radiation-related cancers (HR: 1.59 in males, 1.77 in females). In contrast, the study reported no increased risk for radiation-insensitive cancers (HR: 0.59 in males, 0.98 in females)[35].

Studies in Canada and China found no clear association between proximity to nuclear power plants and cancer incidence. A study in Ontario, Canada, reported no consistent pattern for all cancers combined or for specific cancers such as thyroid, lung, breast, stomach, colon, bladder, brain, liver, leukemia, and non-Hodgkin lymphoma among populations living within 25 km of three nuclear plants[1]. Similarly, a study in China on the Qinshan Nuclear Power Plant found no increased risk for all cancers, leukemia, or thyroid cancers[36].

While differences in nuclear power plant technology and emission controls may contribute to variation across studies, much of the observed heterogeneity in findings likely stems from differences in exposure assessment methods, study designs, statistical power, and

outcome definitions. Many previous studies relied on binary proximity cutoffs rather than incorporating cumulative proximity from multiple plants, which may have introduced measurement error by over-simplifying exposure levels in these ecological studies. In contrast, our study uses a continuous, inverse-distance weighted nuclear power plants proximity metric, allowing for a more refined assessment of long-term exposure gradients.

Additionally, studies that focus on single plants or small geographic areas may have limited statistical power, particularly when assessing rare cancers. These studies may lack sufficient case numbers to detect associations, whereas larger-scale national studies, like ours, have the advantage of higher statistical power and the ability to capture regional nuclear power plants proximity variability.

Our study has several limitations. First, our nuclear power plant proximity exposure assumes equal contribution from all nuclear

power plants within 200 km and does not include direct radiation measurements (dosimetry). Second, we analyzed all cancer types combined, even though different malignancies have varying latency periods and radiation sensitivities. Third, our analysis was done at the county (FIPS) level because that is the resolution at which the CDC provides cancer mortality data. This ecological design does not capture individual-level exposure or outcomes and therefore limits causal inference. Moreover, our exposure metric reflects geographic proximity rather than actual radiation exposure experienced by individuals. Fourth, we did not analyze childhood cancers. These outcomes were rare in our dataset, and stratification by age group, sex, county, and year resulted in sparse data and unstable estimates. Proper evaluation of childhood cancer risk would require different modeling approaches tailored to rare outcomes. Fifth, we used a standard formula for calculating the attributable fraction (AF), which assumes a causal relationship between exposure and outcome and does not account for potential unmeasured confounding or exposure misclassification. Finally, our analysis does not incorporate residential histories, and exposure assignment was based on the geographic centroid of each county. While this limits temporal precision compared to individual-level cohort studies, there is no evidence that residential mobility is systematically related to proximity to nuclear plants. As such, any exposure measurement error is likely non-differential, which would bias results toward the null. Our study spans nearly two decades, and we also assume long-term residence stability throughout this period. Despite these limitations, to our knowledge, this is the only national study in the U.S. to examine nuclear power plant proximity and cancer mortality using a continuous proximity metric. Unlike prior studies that focused on single plants, small geographic areas or relied on fixed distance cutoffs to assign binary exposures, our approach provides a comprehensive and a continuous assessment of nuclear power plants proximity across the entire country, capturing the cumulative impact of multiple facilities over nearly two decades and enhancing statistical power to detect potential associations. Additionally, we used a continuous, inverse-distance weighted proximity metric instead of categorical proximity-based definitions, allowing for a more nuanced and comprehensive exposure assessment.

We also utilized a nationally representative, long-term cancer mortality dataset from the CDC, ensuring high data completeness and broad geographic coverage. The inclusion of a full 19 years of cancer mortality data (2000–2018) and a 10-year average nuclear power plants proximity window allows for a robust temporal assessment of long-term proximity effects.

We found that U.S. counties located closer to operational nuclear power plants experienced higher cancer mortality rates, with the strongest associations observed in older adults, particularly among males aged 65–74 and females aged 55–64. These results indicate a spatial association between residential proximity to nuclear power plants and cancer mortality. This study focused exclusively on cancer mortality and did not examine neurological, cardiovascular, or other potential health outcomes associated with nuclear facilities. While current evidence remains insufficient to draw definitive causal conclusions regarding cancer risks among populations living near nuclear plants, our findings highlight an important area for future investigation. Understanding the potential long-term health implications of nuclear power generation is particularly important given the renewed interest in nuclear energy as a low-carbon solution, emphasizing the importance of addressing these potentially substantial but overlooked risks to public health.

## Methods

This research complies with all relevant ethical regulations. The study protocol was reviewed by the Institutional Review Board of the Harvard T.H. Chan School of Public Health (Ethics Board Registration Number [FWA]: FWA00002642; Study Protocol Number: IRB24-0094), which determined that the project does not involve human subjects research as defined by U.S. Department of Health and Human Services and U.S. Food and Drug Administration regulations.

### Nuclear power plants data

Data on the locations and years of nuclear power plants were obtained from the U.S. Energy Information Administration (EIA) website. In addition to U.S. plants, facilities located outside the U.S. (Canada) but within 200 km of a center of a U.S. County were also included in the analysis. When available, plant-specific websites were used to verify and validate locations and operational details. Supplementary Dataset 1 provides a comprehensive list of all included plants along with their operational years, while Fig. 5 illustrates their geographic distribution.

### Cancer mortality data

Cancer mortality data for the contiguous U.S. from 2000 to 2018 were obtained from the Centers for Disease Control and Prevention (CDC). The dataset includes individual mortality records for all deaths, from which we filtered cancer deaths as those with an ICD-10 code beginning with 'C', covering all malignant neoplasms. Unlike the publicly available CDC Wonder database, this dataset contains uncensored mortality records, ensuring full data coverage for all counties.

Our analysis focused on cancer mortality across six age groups (35–44, 45–54, 55–64, 65–74, 75–84, and 85+), stratified by sex (as per the CDC files assignment). The total number of cancer deaths across age groups and gender is summarized in Table S1 of the Supplementary Information.

### Covariates data

We controlled for selected annual county-level covariates (2000–2018) that could potentially confound the association between nuclear power plants proximity and cancer or independently relate to cancer. These covariates include educational attainment, median household income, poverty level, racial composition (White, Asian, African American), population density, temperature, relative humidity, current smoking prevalence, mean BMI, proximity to the nearest hospital, age over 65, percent of persons over 65 with ambulatory physician visits each year, and renting percentage, as detailed in Supplementary Tables S2.

### Nuclear power plants proximity assessment

To estimate nuclear power plants proximity at the county level, we used a proximity-based approach, calculating the inverse of the distance $(1/d)$ in kilometers for each unique plant-county combination, considering all plants that were operational for at least one year between 1990 and 2018 and located within 200 km of a county centroid.

For each plant-county pair, we first computed $1/d$ for every year in which the plant was active. From the year of decommissioning onward, we assigned a value of 0 for $1/d$. We then applied a 10-year average proximity to each individual plant's $1/d$ values to account for cumulative proximity and latency effects. Finally, for each county, we summed the average proximity $1/d$ values across all contributing plants, generating an annual aggregate proximity estimate at the county level.

Figure 5 illustrates the geographic distribution of all nuclear power plants and the counties included in the study, while Supplementary Dataset 1 lists each plant along with its operational years.

This approach allows for a continuous assessment of long-term proximity trends, ensuring that past proximity contributions remain relevant for a period even after plant closures, to better align with the latency periods relevant to many adult cancers.

### Statistical analysis

We analyzed yearly cancer mortality data from 2000 to 2018 for all U.S. counties within 200 km of a nuclear power plant,

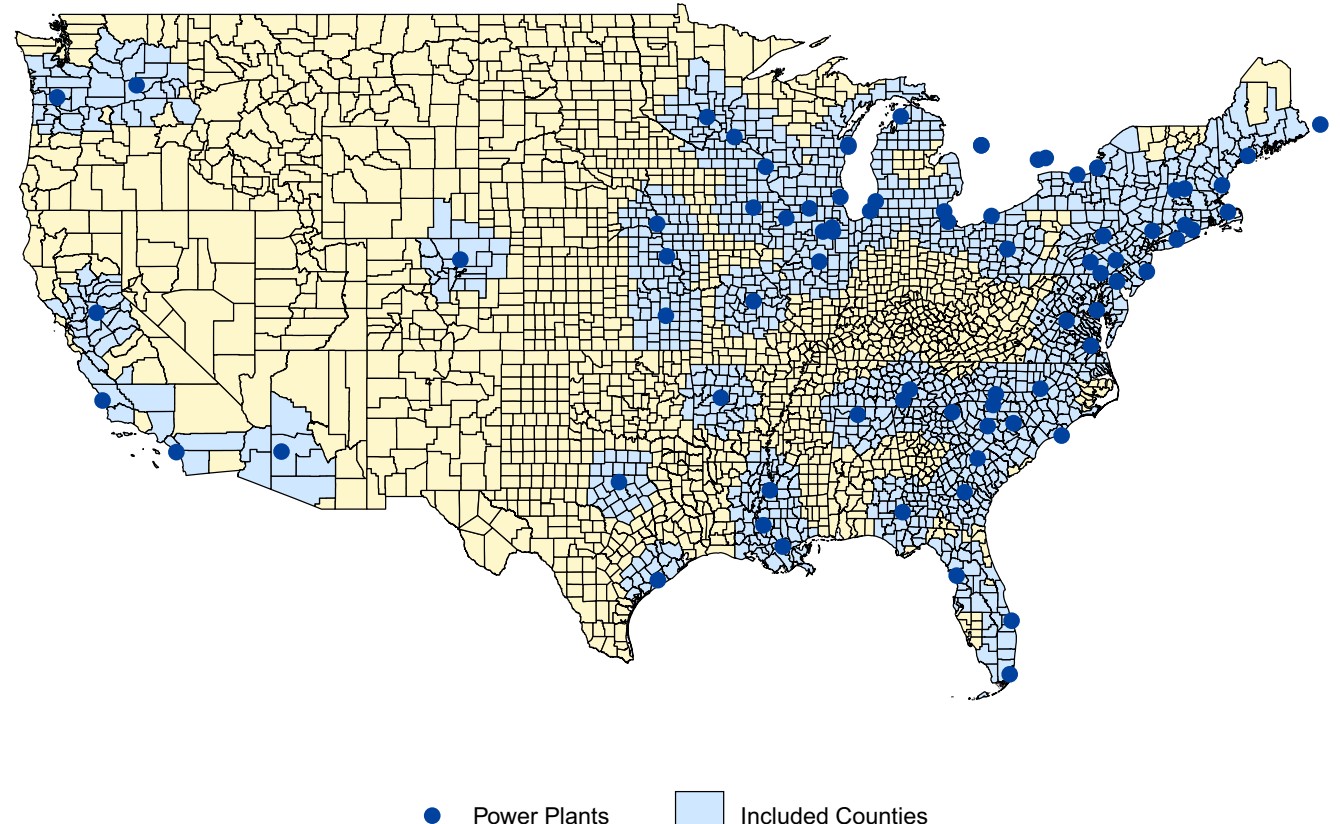

● Power Plants ▢ Included Counties

**Fig. 5 | Geographic distribution of nuclear power plants in the United States and counties within 200 km of a plant operational for at least one year between 1980 and 2018.** Map illustrating the locations of all nuclear power plants (dark blue circles) and U.S. counties located within 200 km of any operational plant (light blue shaded areas) during the study period. County boundaries are outlined in black. *Source data are provided as a Source Data file.*

assessing the relationship between nuclear power plants' proximity and cancer mortality. We used a Generalized Estimating Equation (GEE) Poisson regression model (Eq. 1) to estimate the association between the proximity to nuclear power plants and cancer mortality, while adjusting for relevant county-level covariates. The list of covariates is detailed in Supplementary Table S2.

The model's outcome variable was the total number of cancer deaths per year per county, stratified by sex and age group. To account for population size differences across counties and demographic groups, we included a natural log population offset (specific to each age group, county, sex, and year), allowing the model to estimate mortality rates rather than counts. This ensures that model coefficients are interpretable on the rate scale. The model used county as the clustering variable to account for within-cluster correlations over time, assuming an exchangeable correlation structure, where all observations within a county have the same correlation. Additionally, robust (sandwich) standard errors were used to ensure valid inference by adjusting for potential model misspecification and within-county correlation.

$$\log(\lambda_{ijk}) = \beta_0 + \beta_1 \cdot C1_{ij} + \beta_2 \cdot C2_{ij} + \beta_3 \cdot C3_{ij} + \beta_4 \cdot C4_{ij} + \cdots + \beta_p \cdot Cp_{ij}$$
$$+ \beta_e E_{ij} + \text{og}(\text{population}_{ijk}) \quad (1)$$

where:
- $i$ indexes counties, $j$ indexes years, and $k$ indexes age–sex groups.
- $\lambda_{ijk}$ represents the expected number of cancer deaths in county $i$ at year $j$ for age group $k$.
- $C_{nij}$ denotes covariate $n$ for county $i$ and year $j$.

- $E_{ij}$ represents the sum (over plants) of the inverse-distance proximity metric, and $\beta_e$ is its estimated effect.
- $\beta_O$ is the intercept, representing the baseline log cancer mortality rate.

We stratified our analysis by sex and modeled mortality for the six age groups. For each sex-age combination, we ran separate GEE Poisson regression models to estimate the association between nuclear power plants' proximity and cancer mortality.

The yearly relative risks (RRs) for cancer mortality were calculated for each county, age group, sex, and year using Eq. (2). These RRs were used to estimate attributable mortality fractions (AFs) using Eq. (3).

To compute the total number of cancer deaths attributable to nuclear power plants' proximity during the study period (2000–2018), we first calculated yearly attributable cases separately for each county, age group, sex, and year. Next, we summed the attributable cancer deaths across all counties and years for each age group and sex. The results present the total number of attributable cancer deaths for each age group and sex, aggregated over the entire study duration.

$$RR = e^{(Proximity * Coefficient)} \quad (2)$$

$$AF = \frac{RR - 1}{RR} \quad (3)$$

## Statistics and reproducibility
All analyses were performed using RStudio (version 2023.09.0 + 463) with R (version 4.3.2). Statistical modeling, data processing, and visualization were conducted using the following R packages:

geepack[37] for generalized estimating equation (GEE) models, data.table[38] and dplyr[39] for data manipulation, sf[40] and raster[41] for spatial operations, and ggplot2[42] for visualization.

The study design was observational and ecological, based on population-level data aggregated by county, year, age group, and sex. The statistical unit of analysis was the county–year–age–sex group.

No statistical method was used to predetermine sample size. The analysis included all available observations meeting inclusion criteria (counties located within 200 km of an operational nuclear power plant with complete covariate data for 2000–2018).

Given the nature of the study design, randomization and blinding were not applicable. All analyses were conducted using pre-specified modeling approaches and reproducible R scripts.

The reproducibility of results was confirmed through independent reruns of the statistical code and sensitivity analyses using alternative proximity definitions and tapering periods, which produced consistent findings.

### Reporting summary
Further information on research design is available in the Nature Portfolio Reporting Summary linked to this article.

## Data availability
The cancer mortality data used in this study were obtained from the Centers for Disease Control and Prevention (CDC) and are confidential; therefore, they cannot be publicly shared. Researchers may request access directly from the CDC. All other data sources used in this analysis are publicly available. The aggregated data underlying all main figures and tables are provided as a Source Data file. Source data are provided with this paper.

## Code availability
The R code used to perform the statistical and epidemiological analyses is available from the corresponding author upon reasonable request.

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

## Acknowledgements

No grants or external funding were awarded for this work. Y.A. acknowledges support from the Harvard T.H. Chan School of Public Health through a doctoral fellowship.

## Author contributions

Y.A. performed the literature review, designed the exposure metric, curated the data, developed the epidemiologic model, conducted the statistical analyses, and drafted and revised the manuscript. J.S.E. contributed to the study's conceptualization, design, and manuscript writing. B.C. and J.S. provided statistical guidance and participated in the methodology. P.K. served as the senior author and advisor on all steps. B.A., C.L.Z.V., P.J.L., D.C.C., E.G., and M.K. contributed to conceptualization and provided critical feedback on the initial draft. All authors reviewed and approved the final manuscript.

## Competing interests

The authors declare no competing interests.
