## [Transparent Peer Review file · Nature Communications]

National Analysis of Cancer Mortality and Proximity to Nuclear Power Plants in the United States

Corresponding Author: Dr Yazan Alwadi

Version 0:

Reviewer comments:

Reviewer #1

(Remarks to the Author)

“A National Analysis of the Impact of Nuclear Power Plants on All-Cancer Mortality in the U.S. During 2000–2018”

The authors examine whether the distance between a county’s geographic centroid and neighboring nuclear power plants is associated with the county-level all cancer mortality rates. The analyses were conducted with strata of sex and age group, with results aggregated to summarize results. The paper reports evidence of positive associations between county-level proximity to nuclear power plants and annual cancer mortality rates aggregated at the county level.

Given the contemporary context in the United States, with strong federal support for efforts to restart formerly decommissioned reactors, relicense aging reactors currently in operation and scheduled for shutdown, and license new reactors, the paper should be of interest to the journals readers.

However, the authors should do more to acknowledge limitations of this work. Below I note several major limitations, as well as a few minor editorial suggestions.

1. Age restriction. Analyses were restricted to cancers occurring at adult ages. The authors should clarify that childhood cancer was not examined. This fact is omitted in the abstract, Introduction, and not mentioned in the Discussion. Moreover, some of the prior studies cited as supportive of an association (e.g., the German study by Spix et al.) are focused solely on childhood cancer. A bit of discussion regarding why childhood cancers were not examined here (perhaps data access issues?) and the limitations of their exclusion, given steeper dose-response coefficients for childhood than adult exposures in many studies, fewer confounders, and shorter latency periods. A major limitation of the study is lack of data on childhood cancers and this shouldn’t go unmentioned.
2. The authors should be careful to distinguish between: a) distance between a county’s geographic centroid neighboring nuclear power plants; and, b) radiation exposure from nuclear plants. The authors write that proximity is “a relative measure of nuclear power plant-related radiation exposure” which is a questionable assertion. Consider an ingestion pathway: radionuclides contaminate fruit trees, and the fruit is driven to market. Is $1/d$ a relative measure of exposure in this case? It is appropriate to frame interest in this question as motivated by concerns about environmental releases of radiation from a plant; however, it is unclear whether radiation from the plant is what is being studied or whether distance is a proxy for other factors. Language such as “association between nuclear exposure and cancer mortality” conveys the impression that what is being studied is radiation rather than proximity. Certainly, official estimates of offsite releases suggest very low doses even at fence line distances, such that no observable effect would be expected based on standard radiation risk models. A major limitation of the study is lack of dosimetry and this shouldn’t be downplayed.
3. The authors model distance via a $1/d$ transformation of distance, with no description of exploration of the shape of this association. One could imagine other plausible models for distance-mortality associations: d , $1/d^2$, etc. Discuss exploration or sensitivity to this transformation. The only presentation of results are of summary outcomes under this assumed parametric function (as I understand it, linear in $1/d$). It would be useful to describe or illustrate the appropriateness of this model (does the rate increase linearly as a function of $1/d$?).
4. Covariate selection. Some of the covariates included are surprising: humidity level, for example. Discuss the selection of

covariates and logic. Were these evaluated for contribution to model fit or was variable selection done a priori?

5. Outcome definition. The analyses examine the broad outcome category of all cancer mortality (i.e., all deaths with ICD 10 codes commencing with C). Explain choice of this outcome, and why specific cancer sites were not examined. Arguably, greater insight could be gained by examination of specific cancer outcomes.

Minor

1. Moving average versus exposure window. The authors write “We then applied a 10-year moving average to each individual plant’s 1/d values to account for cumulative exposure and latency effects.” I found this confusing. For county i at year j , I believe what they are saying is the exposure metric is defined as average exposure in the window $[j-9, j]$. A moving average would suggest that you examined a series of windows in your analysis, such as $[j-10, j-1]$, and $[j-11, j-2]$. I don’t think you explored those options; if so, it would be better to characterize the exposure metric as the cumulative (or average) exposure within a defined exposure window – not as a ‘moving average.’

2. Risk versus rate. Attention to distinction between ‘risk’ and ‘rate’ – While a bit pedantic, I found it difficult to quickly understand the outcome model when I read in the abstract, for example, “estimate relative risks (RR) for cancer mortality.” One could think that the goal was to estimate the risk of cancer over the period 2000-2018, when in fact what was modeled was the annual (sex- age-specific) cancer rate. I suggest revise to read “estimate the relative rate of cancer mortality.”

3. Model offset. The authors write “To account for population size differences across counties and demographic groups, we included a natural log population offset (specific to each age group, county, sex, and year), ensuring that mortality rate predictions were appropriately scaled.” Rather than characterize the offset as a scaling of the rate prediction, it would be clearer to note that by incorporating the log person-time offset, you can obtain a rate prediction (i.e., the estimated model coefficients can be interpreted as describing the cancer rate, rather than count, in the study population).

4. Spatial clustering. The authors used GEE under a Poisson regression model. The model allowed for spatial clustering at the county level (i.e., rates in a county over the years 2000-2018). It would be informative to report the results of a standard Poisson regression, as I suspect the allowance for within county-clustering of age-sex cancer mortality rates over time via an exchangeable correlation structure and robust variance estimator is a lot of model complexity that may be unnecessary (and arguably unjustified) in analyses of mortality outcomes.

5. AF. It would be useful to comment in the Discussion on the limitations of the simple expression for ‘attributable fraction’ used here and the critiques of it (relative to other approaches to estimation of average causal effects).

6. You write “In contrast, our study uses a continuous, inverse-distance weighted exposure metric, allowing for a more refined assessment of long-term exposure gradient (lines 333-335)” which might give the impression that this study does much better in terms of assessing proximity as a metric of exposure. It would be good to put this in context of prior work that has done much more in terms of individual residential histories and locations. While limitations are discussed briefly (lines 340-345) major limitations typically receive a single sentence. Discuss the limitations of your analyses that examine county-based mortality rates relative to some prior studies, such as the German study, that examine individual household address and classify each subject based on the distance from their home to neighboring NPPs. Discuss the limitations of your study which geographic centroid of county of residence at time of death to studies that have collected residence history information. What is the likely impact of residential mobility over the preceding 10 years and might this vary with attained age?

7. Citations. While reference to some key studies in this research area, such as Spix et al., 2008, are made in the text (lines 87-91), none of these studies appear in the cited literature. I assume that this is just an error in the bibliographic software?

Reviewer #2

(Remarks to the Author)

General Comments

This manuscript presents a geographical analysis of cancer mortality related to proximity to nuclear power plants in the United States. The results could be interesting, but the bibliography cited is manifestly incomplete, the presentation of the data is misleading (use of the term ‘nuclear exposure’ without any exposure data), and the discussion of the study’s limitations is inadequate.

A major weakness of the manuscript is the lack of information and even discussion on dose levels. The authors use the term “nuclear exposure” without any information on exposure levels, which is clearly inadequate. No mention of exposure levels is provided, and discussion of the fact that distance is a poor indicator of releases from nuclear facilities is virtually absent. Some methodological choices are poorly justified: consideration of adults only, whereas historically most of the concerns about potential risks around NPPs concerned children; pertinence of some of the covariates considered; consideration of all cancers together; the justification of an indicator “accounting for cumulative exposure and latency effects” is unclear... Several major references, essential to the subject of the manuscript, are missing: the national study of cancer risk near nuclear power plants in the United States carried out by the NCI; the in-depth analysis of possible approaches to be used in the United States to study cancer risk near nuclear facilities, published 10 years ago by the NAS (see references below). There is little discussion of the limitations of the study (non-homogeneous size of counties in the United States, potential ecological bias, migration, lack of incidence data, lack of exposure data, exclusion of children, etc.).

Specific/minor comments

Several repetitions in the manuscript could be avoided (for example, the age groups are given 4 times).

The format of the reference citations is not consistent throughout the manuscript.

Abstract, page 3, line 53: The numbers of deaths attributable to nuclear plant proximity can't be interpreted without context (for example, the corresponding number of baseline deaths over the same period and area).

Introduction, page 5, line 82: of course, "Populations residing near nuclear power plants may experience chronic exposure to ionizing radiation, which is a well-established risk factor for multiple cancers", but what is also known is that the level of risk is related to the level of dose, and the doses in the vicinity of nuclear power plant are known to be very limited. A discussion on the levels of dose is clearly missing in the manuscript.

Introduction, page 6, line 95: "lacking a comprehensive national assessment". This is wrong, a national study of cancer risk near NPPs in the United States has been published in the 1990's (see references below).

Exposure assessment, page 9, line 153: Sentence "This approach allows for a more realistic assessment of long-term exposure trends" is exaggerated.

Exposure assessment, page 9, line 155: Sentence "aligning with the latency periods of radiation-related cancers" is also exaggerated. The authors consider all cancer mortality and not radiation-related cancers. Furthermore, the latency between radiation risk and cancer risk varies according to cancer site.

Exposures, page 11, figure 2: the unit is missing, and the interpretation of the figure is unclear

Discussion, page 17, line 275: "This pattern aligns with the known age-related increase in cancer risk, potentially driven by cumulative radiation exposure and biological susceptibility in older populations". This formulation is obviously an over-interpretation of the results.

Discussion, page 20, line 347: Sentence "to our knowledge, this is the first national study in the U.S. to examine nuclear power plant exposure and cancer mortality". This is wrong, see references "Jablon et al" below.

Additional references

- Jablon, S., Hrubec, Z., et al. (1991). "Cancer in populations living near nuclear facilities. A survey of mortality nationwide and incidence in two states." JAMA 265(11), 1403-8.
- Jablon, S., Hrubec, Z., et al. (1990a). Cancer in populations living near nuclear facilities. Vol 1 - Report and summary. Washington, National Cancer Institute.
- Jablon, S., Hrubec, Z., et al. (1990b). Cancer in populations living near nuclear facilities. Vol 2 - Individual facilities: cancer before and after start-up. Washington, National Cancer Institute.
- Jablon, S., Hrubec, Z., et al. (1990c). Cancer in populations living near nuclear facilities. Vol 3 - Individual facilities: cancer by 5-year time intervals. Washington, National Cancer Institute.
- Committee on the Analysis of Cancer Risks in Populations near Nuclear Facilities (2012) Analysis of Cancer Risks in Populations near Nuclear Facilities—Phase I (Washington, DC: National Research Council (NRC); Nuclear and Radiation Studies Board; Division on Earth and Life Studies)
- Committee on the Analysis of Cancer Risks in Populations Near Nuclear Facilities (2014) Phase 2 Pilot Planning; Nuclear and Radiation Studies Board; Division on Earth and Life Studies; National Research Council. 2014. ISBN 978-0-309-31335-3

Version 1:

Reviewer comments:

Reviewer #1

(Remarks to the Author)

In this revised version of the manuscript, the authors have addressed the primary comments raised in the initial review. They do a better job of distinguishing between a study of proximity to nuclear plants and a study of radiation exposure from nuclear plants (this study does not address the latter).

The discussion provides a fuller discussion of the current study's limitations.

The Discussion provides a fairly limited discussion of prior research on this topic (leaning towards emphasis on studies that have reported evidence of association).

Overall this remains a topic of substantial public interest and this study, despite its limitations, is timely and points to directions for further research.

Reviewers' comments:

Reviewer #1 (Remarks to the Author):

“A National Analysis of the Impact of Nuclear Power Plants on All-Cancer Mortality in the U.S. During 2000–2018”

The authors examine whether the distance between a county’s geographic centroid and neighboring nuclear power plants is associated with the county-level all cancer mortality rates. The analyses were conducted with strata of sex and age group, with results aggregated to summarize results. The paper reports evidence of positive associations between county-level proximity to nuclear power plants and annual cancer mortality rates aggregated at the county level.

Given the contemporary context in the United States, with strong federal support for efforts to restart formerly decommissioned reactors, relicense aging reactors currently in operation and scheduled for shutdown, and license new reactors, the paper should be of interest to the journal’s readers.

However, the authors should do more to acknowledge limitations of this work. Below I note several major limitations, as well as a few minor editorial suggestions.

1. Age restriction. Analyses were restricted to cancers occurring at adult ages. The authors should clarify that childhood cancer was not examined. This fact is omitted in the abstract, Introduction, and not mentioned in the Discussion. Moreover, some of the prior studies cited as supportive of an association (e.g., the German study by Spix et al.) are focused solely on childhood cancer. A bit of discussion regarding why childhood cancers were not examined here (perhaps data access issues?) and the limitations of their exclusion, given steeper dose-response coefficients for childhood than adult exposures in many studies, fewer confounders, and shorter latency periods. A major limitation of the study is lack of data on childhood cancers and this shouldn’t go unmentioned.

We thank the reviewer for highlighting this important point. We agree that the exclusion of childhood cancers should be explicitly acknowledged in the manuscript. In our analyses, we restricted to adult cancers due to the extremely small number of cancer deaths in younger age groups. For example, between ages 0–14, fewer than 20,000 deaths were observed nationally over the study period (in studied counties), compared to over 6 million among adults. When stratifying by age group, sex, year, and county, the rarity of childhood cancer deaths resulted in highly sparse data and statistically unstable estimates. Accurately modeling these outcomes would require a different methodological approach and was beyond the scope of the current study.

We have now clarified this point in the limitations section of the discussion.

Age Group	Total Cancer Deaths
85+	1,229,433
75_84	2,115,813
65_74	1,941,033
55_64	1,371,020
45_54	618,516
35_44	171,201
25_34	48,034
15_24	21,017
5_14	12,343
1_4	4,912
0-1	996

2. The authors should be careful to distinguish between: a) distance between a county's geographic centroid neighboring nuclear power plants; and, b) radiation exposure from nuclear plants. The authors write that proximity is "a relative measure of nuclear power plant-related radiation exposure" which is a questionable assertion. Consider an ingestion pathway: radionuclides contaminate fruit trees, and the fruit is driven to market. Is $1/d$ a relative measure of exposure in this case? It is appropriate to frame interest in this question as motivated by concerns about environmental releases of radiation from a plant; however, it is unclear whether radiation from the plant is what is being studied or whether distance is a proxy for other factors. Language such as "association between nuclear exposure and cancer mortality" conveys the impression that what is being studied is radiation rather than proximity. Certainly, official estimates of offsite releases suggest very low doses even at fence line distances, such that no observable effect would be expected based on standard radiation risk models. A major limitation of the study is lack of dosimetry and this shouldn't be downplayed.

We thank the reviewer for this insightful and important comment. In response, we have revised the manuscript language throughout to replace terms such as "nuclear exposure" with "proximity to nuclear power plants" to more accurately reflect the nature of our exposure metric. We agree that our use of inverse distance ($1/d$) does not represent radiation dosimetry and should not be interpreted as a direct measure of radiation exposure.

Rather, we used $1/d$ as a first-order proxy for spatial proximity to nuclear power plants, acknowledging that this metric may indirectly reflect various potential exposure pathways

(e.g., air, water, food chain) but does not capture them with specificity or precision. We selected this simplified metric as a starting point for national-scale ecological analysis, recognizing that designing a more mechanistically accurate model for one exposure pathway (e.g., ingestion) may compromise performance for others or require data not available at this scale.

We now explicitly discuss in the limitations that our metric does not reflect actual radiation dose, and that lack of individual-level dosimetry is a major limitation of this study. We have added language to emphasize that our findings should be interpreted as associations with proximity, not with quantified radiation exposure.

3. The authors model distance via a $1/d$ transformation of distance, with no description of exploration of the shape of this association. One could imagine other plausible models for distance-mortality associations: d , $1/d^2$, etc. Discuss exploration or sensitivity to this transformation. The only presentation of results are of summary outcomes under this assumed parametric function (as I understand it, linear in $1/d$). It would be useful to describe or illustrate the appropriateness of this model (does the rate increase linearly as a function of $1/d$?).

We thank the reviewer for this insightful suggestion. In response, we explored alternative transformations, including $1/d^2$ and $1/d^3$, to evaluate the sensitivity of our results to this modeling assumption. However, both $1/d^2$ and $1/d^3$ provided a notably worse fit (higher QIC for all age groups with significant results) to the observed cancer mortality patterns across age and sex groups, as illustrated in the figure and table below (for $1/d^2$). The $1/d$ transformation best aligned with the observed data and provided the most stable and interpretable estimates. For these reasons, and to maintain clarity in the manuscript, we have retained the $1/d$ specification and have not included these sensitivity results in the main text. We are happy to share additional plots, tables or include these sensitivities if the reviewer feels it is necessary.

Age_Group	Gender	QIC_1overd	QIC_1overd2	QIC_Diff	Better_Model
35_44	Male	-172761	-172766	-5.116	1/d^2 better
35_44	Female	-295288	-295290	-1.397	1/d^2 better
45_54	Male	-1531515	-1531511	4.111	1/d better
45_54	Female	-1614683	-1614667	16.272	1/d better
55_64	Male	-4974203	-4974125	78.413	1/d better
55_64	Female	-4033444	-4033358	85.072	1/d better
65_74	Male	-7757596	-7757420	175.8	1/d better
65_74	Female	-6227355	-6227267	87.524	1/d better
75_84	Male	-8311752	-8311609	143.048	1/d better
75_84	Female	-7568649	-7568580	68.916	1/d better
85+	Male	-3569510	-3569478	32.462	1/d better
85+	Female	-4545949	-4545894	54.865	1/d better

4. Covariate selection. Some of the covariates included are surprising: humidity level, for example. Discuss the selection of covariates and logic. Were these evaluated for contribution to model fit or was variable selection done a priori?

We thank the reviewer for raising this important point. Our covariate selection strategy was grounded in both theoretical considerations and prior literature, with the goal of identifying a minimal adjustment set that would account for potential confounders—variables that might be associated both with proximity to nuclear power plants and cancer mortality rates—while avoiding inclusion of mediators on the causal pathway.

Specifically:

- Socio-demographic covariates were included based on evidence linking them to both cancer risk and residential location patterns.
- Health behavior-related covariates were included due to their relationship with cancer mortality, and their spatial variation across counties.
- Environmental and meteorological covariates, including temperature (tmean), relative humidity (RH), and population density, were incorporated to adjust for factors that may influence airborne pollutant transport and dispersion.
- Year was included as a categorical variable to control for trends in cancer mortality.

We acknowledge that RH and temperature may be less familiar in cancer mortality models, but we note their relevance to the behavior of environmental exposures such as particulate matter and radiation-related particles. As a sensitivity analysis, we re-estimated our models after removing RH and tmean and observed that the estimated coefficients for nuclear proximity remained consistent in both direction and magnitude, as shown in the plot below. This confirms that our results are not dependent on the inclusion of these environmental covariates.

We recognize that not all included covariates are classic confounders. However, none are hypothesized to be mediators of the relationship between proximity to nuclear power plants and cancer mortality.

5. Outcome definition. The analyses examine the broad outcome category of all cancer mortality (i.e., all deaths with ICD 10 codes commencing with C). Explain choice of this outcome, and why specific cancer sites were not examined. Arguably, greater insight could be gained by examination of specific cancer outcomes.

We thank the reviewer for this important observation. Our choice to examine all cancer mortality (ICD-10 codes beginning with “C”) was driven by the need to ensure sufficient case counts within each county-year-age-sex stratum. Even after aggregating across all cancer types, strata in younger age groups yielded very low counts, leading to unstable rate estimates that are not suitable for reliable inference using our chosen modeling framework.

Our approach requires a minimum number of cases to generate stable and interpretable estimates within this highly stratified spatiotemporal design. As shown in the table below, cancer mortality in younger age groups remains limited in absolute numbers, even at the national scale:

Age Group	Total Cancer Deaths
85+	1,229,433
75_84	2,115,813
65_74	1,941,033
55_64	1,371,020
45_54	618,516
35_44	171,201
25_34	48,034
15_24	21,017
5_14	12,343
1_4	4,912
0-1	996

We fully agree that analyses of site-specific cancers could provide more mechanistic insight. We are currently conducting complementary studies using both cancer mortality and incidence data at ZIP code and county levels to examine site-specific outcomes (e.g., breast, prostate, colorectal, thyroid). These studies, tailored to data availability and outcome rarity, use alternative modeling strategies and are currently under review or in preparation. We have shared select results from these investigations with the Editor to provide additional context and demonstrate the consistency of findings. Notably, their results are aligned with the broader trends reported in the present manuscript.

Minor

1. Moving average versus exposure window. The authors write “We then applied a 10-year moving average to each individual plant’s 1/d values to account for cumulative exposure and latency effects.” I found this confusing. For county i at year j , I believe what

they are saying is the exposure metric is defined as average exposure in the window [j-9, j]. A moving average would suggest that you examined a series of windows in your analysis, such as [j-10, j-1], and [j-11, j-2]. I don't think you explored those options; if so, it would be better to characterize the exposure metric as the cumulative (or average) exposure within a defined exposure window – not as a 'moving average.'

We thank the reviewer for bringing our attention to this point, we have now changed the wording of “moving average” to “average proximity”.

2. Risk versus rate. Attention to distinction between 'risk' and 'rate' – While a bit pedantic, I found it difficult to quickly understand the outcome model when I read in the abstract, for example, “estimate relative risks (RR) for cancer mortality.” One could think that the goal was to estimate the risk of cancer over the period 2000-2018, when in fact what was modeled was the annual (sex- age-specific) cancer rate. I suggest revise to read “estimate the relative rate of cancer mortality.”

We thank the reviewer for this comment; we have now revised the wording in the abstract accordingly.

3. Model offset. The authors write “To account for population size differences across counties and demographic groups, we included a natural log population offset (specific to each age group, county, sex, and year), ensuring that mortality rate predictions were appropriately scaled.” Rather than characterize the offset as a scaling of the rate prediction, it would be clearer to note that by incorporating the log person-time offset, you can obtain a rate prediction (i.e., the estimated model coefficients can be interpreted as describing the cancer rate, rather than count, in the study population).

We thank the reviewer for this helpful clarification. We agree that describing the offset in terms of enabling estimation of rates, rather than scaling counts, is more accurate. We have revised the language in the manuscript accordingly to reflect that the log population offset allows for the interpretation of model coefficients as describing cancer mortality rates rather than counts.

4. Spatial clustering. The authors used GEE under a Poisson regression model. The model allowed for spatial clustering at the county level (i.e., rates in a county over the years 2000-2018). It would be informative to report the results of a standard Poisson regression, as I suspect the allowance for within county-clustering of age-sex cancer mortality rates over time via an exchangeable correlation structure and robust variance estimator is a lot of model complexity that may be unnecessary (and arguably unjustified) in analyses of mortality outcomes.

We thank the reviewer for this insightful comment. As suggested, we reran the analyses using a standard Poisson regression without accounting for within-county clustering or correlation structure. The results are presented below. As shown, the estimates remain consistent and robust, supporting the validity of our main findings. This consistency suggests that our conclusions are not driven by model complexity, and that the additional structure in the GEE model did not materially alter the interpretation of the exposure effect.

5. AF. It would be useful to comment in the Discussion on the limitations of the simple expression for ‘attributable fraction’ used here and the critiques of it (relative to other approaches to estimation of average causal effects).

We thank the reviewer for this important point. We have now discussed this limitation in the Discussion highlighting the standard attributable fraction (AF) formula we used. Specifically, we note that it assumes a causal relationship between exposure and outcome and does not account for potential unmeasured confounding or exposure misclassification.

6. You write “In contrast, our study uses a continuous, inverse-distance weighted exposure metric, allowing for a more refined assessment of long-term exposure gradient (lines 333-335)” which might give the impression that this study does much better in terms of assessing proximity as a metric of exposure. It would be good to put this in context of prior work that has done much more in terms of individual residential histories and locations. While limitations are discussed briefly (lines 340-345) major limitations typically receive a single sentence. Discuss the limitations of your analyses that examine county-

based mortality rates relative to some prior studies, such as the German study, that examine individual household address and classify each subject based on the distance from their home to neighboring NPPs. Discuss the limitations of your study which geographic centroid of county of residence at time of death to studies that have collected residence history information. What is the likely impact of residential mobility over the preceding 10 years and might this vary with attained age?

We thank the reviewer for this important point. We have revised the limitations section to explicitly clarify that our study is ecological and cannot match the spatial precision of individual-level cohort studies. We also clarified that we compared our study to other ecological designs. We further note that while residential mobility could introduce exposure measurement error, there is no evidence that people relocate based on perceived nuclear risk, especially given that nuclear plant operation is not widely associated with health harms in public discourse. Therefore, we expect any measurement error to be non-differential, which would bias our findings toward the null. We now include this discussion in the manuscript to better contextualize our results.

7. Citations. While reference to some key studies in this research area, such as Spix et al., 2008, are made in the text (lines 87-91), none of these studies appear in the cited literature. I assume that this is just an error in the bibliographic software?

We thank the reviewer for pointing this out. As noted, this was indeed an issue with the bibliographic software, and we have now corrected the references to ensure that all cited studies are properly listed in the bibliography.

Reviewer #2 (Remarks to the Author):

General Comments

This manuscript presents a geographical analysis of cancer mortality related to proximity to nuclear power plants in the United States. The results could be interesting, but the bibliography cited is manifestly incomplete, the presentation of the data is misleading (use of the term 'nuclear exposure' without any exposure data), and the discussion of the study's limitations is inadequate.

A major weakness of the manuscript is the lack of information and even discussion on dose levels. The authors use the term "nuclear exposure" without any information on

exposure levels, which is clearly inadequate. No mention of exposure levels is provided, and discussion of the fact that distance is a poor indicator of releases from nuclear facilities is virtually absent.

We thank the reviewer for highlighting these important points. As noted by Reviewer 1, there was a technical issue with the bibliographic software that has now been resolved; all relevant citations now appear in the reference list. We have revised the manuscript to replace the term “nuclear exposure” with “proximity to nuclear power plants” to more accurately describe our exposure metric. Additionally, we have substantially expanded the limitations section to address concerns regarding ecological inference, lack of dosimetry, individual-level exposure assessment, and other methodological constraints.

Some methodological choices are poorly justified: consideration of adults only, whereas historically most of the concerns about potential risks around NPPs concerned children; pertinence of some of the covariates considered; consideration of all cancers together; the justification of an indicator “accounting for cumulative exposure and latency effects” is unclear...

We thank the reviewer for this important observation. Our choice to examine all cancer mortality (ICD-10 codes beginning with “C”) was driven by the need to ensure sufficient case counts within each county-year-age-sex stratum. Even after aggregating across all cancer types, strata in younger age groups yielded very low counts, leading to unstable rate estimates that are not suitable for reliable inference using our chosen modeling framework.

Our approach requires a minimum number of cases to generate stable and interpretable estimates within this highly stratified spatiotemporal design. As shown in the table below, cancer mortality in younger age groups remains limited in absolute numbers, even at the national scale:

Age Group	Total Cancer Deaths
85+	1,229,433
75_84	2,115,813
65_74	1,941,033
55_64	1,371,020
45_54	618,516
35_44	171,201
25_34	48,034
15_24	21,017
5_14	12,343

1_4	4,912
0-1	996

Our covariate selection strategy was grounded in both theoretical considerations and prior literature, with the goal of identifying a minimal adjustment set that would account for potential confounders (variables that might be associated both with proximity to nuclear power plants and cancer mortality rates) while avoiding inclusion of mediators on the causal pathway.

Specifically:

- Socio-demographic covariates were included based on evidence linking them to both cancer risk and residential location patterns.
- Health behavior-related covariates were included due to their relationship with cancer mortality, and their spatial variation across counties.
- Environmental and meteorological covariates, including temperature (tmean), relative humidity (RH), and population density, were incorporated to adjust for factors that may influence airborne pollutant transport and dispersion.
- Year was included as a categorical variable to control for trends in cancer mortality.

We acknowledge that RH and temperature may be less familiar in cancer mortality models, but we note their relevance to the behavior of environmental exposures such as particulate matter and radiation-related particles. As a sensitivity analysis, we re-estimated our models after removing RH and tmean and observed that the estimated coefficients for nuclear proximity remained consistent in both direction and magnitude, as shown in the plot below. This confirms that our results are not dependent on the inclusion of these environmental covariates.

We recognize that not all included covariates are classic confounders. However, none are hypothesized to be mediators of the relationship between proximity to nuclear power plants and cancer mortality. While GEE models are not strictly collapsible, inclusion of non-mediating covariates is still appropriate, as it can improve precision and adjust for residual variation without introducing bias to the estimated population-averaged effect of nuclear proximity.

The phrase “accounting for cumulative exposure and latency effects” refers to two design features of our proximity metric. First, by summing inverse-distance contributions from all nuclear power plants within 200 km, the metric incorporates cumulative proximities from multiple potential sources rather than relying on a binary or single-plant proximity. Second, we applied a 10-year moving average of this metric to reflect latency (the time between exposure and cancer mortality), based on established knowledge that radiation-related cancers often develop over extended periods.

Several major references, essential to the subject of the manuscript, are missing: the national study of cancer risk near nuclear power plants in the United States carried out by the NCI; the in-depth analysis of possible approaches to be used in the United States to study cancer risk near nuclear facilities, published 10 years ago by the NAS (see references below).

There is little discussion of the limitations of the study (non-homogeneous size of counties in the United States, potential ecological bias, migration, lack of incidence data, lack of exposure data, exclusion of children, etc.).

We thank the reviewer for highlighting these important points, we have now expanded to the limitations section to discuss these points.

Specific/minor comments

Several repetitions in the manuscript could be avoided (for example, the age groups are given 4 times).

We thank the reviewer for bringing this to our attention, we have now removed the repetitions.

The format of the reference citations is not consistent throughout the manuscript.

We thank the reviewer for bringing this to our attention, this have now been rectified.

Abstract, page 3, line 53: The numbers of deaths attributable to nuclear plant proximity can't be interpreted without context (for example, the corresponding number of baseline deaths over the same period and area).

We thank the reviewer for highlighting this, we have now added the Afs to the abstract for interpretability.

Introduction, page 5, line 82: of course, "Populations residing near nuclear power plants may experience chronic exposure to ionizing radiation, which is a well-established risk factor for multiple cancers", but what is also know is that the level of risk is related to the level of dose, and the doses in the vicinity of nuclear power plant are known to be very limited. A discussion on the levels of dose is clearly missing in the manuscript.

We thank the reviewer for this important clarification. We agree that the level of cancer risk depends on the radiation dose received. In our manuscript, we do not estimate radiation dose or conduct a dose-response analysis. Rather, the cited sentence in the Introduction was intended to establish biological plausibility for investigating cancer risk near nuclear facilities.

To prevent misinterpretation, we have revised the language to clarify that our analysis does not include dose data or dosimetric modeling. Specifically, we now state:

"Populations residing near nuclear power plants may experience low-level chronic exposure to ionizing radiation via environmental release pathways. While our study does not include dosimetry, ionizing radiation is a well-established carcinogen and thus motivates investigation into proximity-based exposure patterns."

Introduction, page 6, line 95: “lacking a comprehensive national assessment”. This is wrong, a national study of cancer risk near NPPs in the United States has been published in the 1990’s (see references below).

We thank the reviewer for highlighting this important point. We have revised the manuscript to clarify that while such national studies exist, they are few in number.

Exposure assessment, page 9, line 153: Sentence “This approach allows for a more realistic assessment of long-term exposure trends” is exaggerated.

We thank the reviewer for the comment, we have now revised it to say “a continuous” instead of “a more realistic”.

Exposure assessment, page 9, line 155: Sentence “aligning with the latency periods of radiation-related cancers” is also exaggerated. The authors consider all cancer mortality and not radiation-related cancers. Furthermore, the latency between radiation risk and cancer risk varies according to cancer site.

We thank the reviewer for highlighting this. we have now toned down the sentence and it now reads: “to better align with latency periods relevant to many adult cancers”

Exposures, page 11, figure 2: the unit is missing, and the interpretation of the figure is unclear

We thank the reviewer for bringing our attention to this point, we have now added the units (1/km) to the figure and the figure title. We have also adjusted the wording of the explanation to better reflect the meaning of the figure.

Discussion, page 17, line 275: “This pattern aligns with the known age-related increase in cancer risk, potentially driven by cumulative radiation exposure and biological susceptibility in older populations”. This formulation is obviously an over-interpretation of the results.

We thank the reviewer for highlighting this, we have now removed this sentence.

Discussion, page 20, line 347: Sentence “to our knowledge, this is the first national study in the U.S. to examine nuclear power plant exposure and cancer mortality”. This is wrong, see references “Jablon et al” below.

We thank the reviewer for highlighting this, we have now updated the sentence, and it reads as below:

“to our knowledge, this is the first national study in the U.S. to examine nuclear power plant proximity and cancer mortality using a continuous proximity metric”

Reviewer #1 (Remarks to the Author):

In this revised version of the manuscript, the authors have addressed the primary comments raised in the initial review. They do a better job of distinguishing between a study of proximity to nuclear plants and a study of radiation exposure from nuclear plants (this study does not address the latter).

The discussion provides a fuller discussion of the current study's limitations.

The Discussion provides a fairly limited discussion of prior research on this topic (leaning towards emphasis on studies that have reported evidence of association).

Overall this remains a topic of substantial public interest and this study, despite its limitations, is timely and points to directions for further research.

Response: We sincerely thank the reviewer for their thoughtful and constructive comments.

In response to the reviewer's remaining suggestion regarding the discussion of prior research, we have further revised the *Discussion* section to provide a more balanced synthesis of the literature. Specifically, we now highlight both studies that reported no significant association between proximity to nuclear power plants and cancer outcomes, as well as those that observed positive associations.

We thank the reviewer for noting that this topic remains of substantial public interest and for recognizing that our study is timely and contributes valuable insights to guide future research on this important issue.